# Reproducible, data-driven characterization of sleep based on brain dynamics and transitions from whole-night fMRI

Fan Nils Yang*, Dante Picchioni, Jacco A de Zwart, Yicun Wang, Peter van Gelderen, Jeff H Duyn*

Advanced MRI Section, Laboratory of Functional and Molecular Imaging, National Institute of Neurological Disorders and Stroke, National Institutes of Health, Bethesda, United States

**\*For correspondence:**
nilsyang106@gmail.com (FNY);
jeff.duyn@nih.gov (JHD)

**Competing interest:** The authors declare that no competing interests exist.

**Abstract** Understanding the function of sleep requires studying the dynamics of brain activity across whole-night sleep and their transitions. However, current gold standard polysomnography (PSG) has limited spatial resolution to track brain activity. Additionally, previous fMRI studies were too short to capture full sleep stages and their cycling. To study whole-brain dynamics and transitions across whole-night sleep, we used an unsupervised learning approach, the Hidden Markov model (HMM), on two-night, 16 hr fMRI recordings of 12 non-sleep-deprived participants who reached all PSG-based sleep stages. This method identified 21 recurring brain states and their transition probabilities, beyond PSG-defined sleep stages. The HMM trained on one night accurately predicted the other, demonstrating unprecedented reproducibility. We also found functionally relevant subdivisions within rapid eye movement (REM) and within non-REM 2 stages. This study provides new insights into brain dynamics and transitions during sleep, aiding our understanding of sleep disorders that impact sleep transitions.

## eLife assessment

This **important** work, leveraging state-of-the-art whole-night sleep EEG-fMRI methods, advances our understanding of the brain states underlying sleep and wakefulness. Despite a small sample size, the authors present **convincing** evidence for substates within N2 and REM sleep stages, with reliable transition structure, supporting the perspective that there are more than the five canonical sleep/wake states.

## Introduction

Given the significant number of people experiencing sleep issues in modern society, there is a growing need for a better understanding of human sleep and its function (*The Lancet, 2022*). Sleep is characterized by relative stationary states, each believed to serve specific functions. To characterize these states, human sleep research has historically classified sleep into a set of stages using PSG (*Berry et al., 2020*; *Rechtschaffen and Kales, 1968*), which combines electroencephalography (EEG) measures of brain activity with several physiological measures. These sleep stages include the progressively deeper sleep stages of N1, N2, and N3 non-rapid eye movement (NREM), as well as REM. Stages are characterized by different patterns of cortical excitability, as a result of varying levels of modulatory neurotransmitters (*Jones, 2020*). Across a full night of sleep, these stages cyclically

alternate, with REM sleep typically occurring 90 min after falling asleep and becoming longer as the night progresses. This cycling is thought to be related to homeostasis and memory consolidation (*Diekelmann and Born, 2010*; *Strauss et al., 2022*). Neuroimaging studies using techniques such as Positron Emission Tomography (PET) and functional MRI (fMRI) have identified unique activity patterns for each PSG stage, contributing to our understanding of sleep's functional role (*Braun et al., 1997*; *Damaraju et al., 2020*; *Picchioni et al., 2013*; *Rué-Queralt et al., 2021*; *Tagliazucchi and Laufs, 2014*; *Tagliazucchi and van Someren, 2017*; *Zhou et al., 2019*).

While these PSG-guided neuroimaging studies provided new information about sleep function, our understanding of brain dynamics is limited by the low temporal resolution of PSG-based sleep scoring rules (i.e. 30 s epochs), low spatial resolution (i.e. limited EEG channels on the scalp), and the subjective visual inspection rules (*Decat et al., 2022*; *Himanen and Hasan, 2000*; *Lambert and Peter-Derex, 2023*). Alternative to PSG-based sleep staging, applying an unsupervised learning method, the HMM (*Stevner et al., 2019*; *Vidaurre et al., 2017*), to sleep fMRI data can objectively model the time series of sleep and infer sleep brain states that recur at different points during sleep. A recent study demonstrated promising results in capturing NREM sleep transitions by applying HMM to relatively short bouts of sleep (<1 hr) fMRI data (*Stevner et al., 2019*). However, because the REM stage typically occurs 90 min after falling asleep and lasts progressively longer over time, capturing brain dynamics associated with sleep cycling requires whole-night data.

In addition, given that studies on sleep stage transitions have shown promising results in diagnoses of various sleep disorders, including narcolepsy (*Christensen et al., 2015*), chronic fatigue syndrome (*Kishi et al., 2011*), and insomnia (*Wei et al., 2017*), it is of great interest to establish an objective and reliable measurement of brain states transitions within and between PSG sleep stages. To achieve this goal, we applied HMM to a unique and extensive dataset of EEG-fMRI concurrent recordings acquired over 8 hr of sleep each night for two consecutive nights (*Moehlman et al., 2019*). This analysis revealed 21 unique brain states, surpassing the number of PSG-defined sleep stages. For potential application in clinical settings, we tested whether our HMM model trained using night 2 data can predict night 1 data. As it turned out, the identified brain states were highly consistent between night 1 and night 2. Furthermore, analyzing the transition probabilities between HMM states revealed a significant subdivision within N2 and within REM sleep stages. This data-driven, PSG-blind analysis of fMRI data provides reproducible brain states and their transition probabilities, potentially serving as a biomarker of sleep transitions in both normal and clinical settings.

## Results
### HMM brain states

To study brain activity representative of the entire Wake-NREM-REM-Wake sleep cycle, we analyzed data from concurrent whole-brain EEG-fMRI measurements on healthy, non-sleep-deprived participants (n=12, age 24±3.5, eight female) over two successive, entire nights of sleep (*Moehlman et al., 2019*). This data was acquired for an independent project that included eight randomly timed acoustical arousals to gauge sleep depth (*Moehlman et al., 2019*). PSG-based sleep staging was conducted by a sleep technologist, utilizing data from EEG, EMG, ECG, and EOG, following the criteria outlined by the AASM (*Berry et al., 2020*).

Following data preprocessing (see Methods section for details), the fMRI time courses from voxels were spatially averaged within each of the 300 regions of interest (ROIs). These ROIs encompassed cortical, subcortical, and cerebellar areas from the Seitzman 300-ROI atlas (*Seitzman et al., 2020*). To ensure consistency and comparability, the ROI time courses were demeaned and variance-normalized for each participant and then concatenated along the temporal dimension. Of note, all 12 participants exhibited at least one complete sleep cycle, encompassing all four sleep stages (N1-3 and REM), during both night 1 and night 2 (*Moehlman et al., 2019*). This uniquely comprehensive dataset provided a robust foundation for our analyses.

The HMM estimated from the night 2 data encompassed a collection of whole-brain states. Each of these states was characterized as a multivariate Gaussian distribution, incorporating two key components: (i) a mean activation distribution, signifying the average activity levels within each ROI when a state was active, and (ii) a functional connectivity (FC) matrix, representing the temporal co-variations among ROIs while in that state.

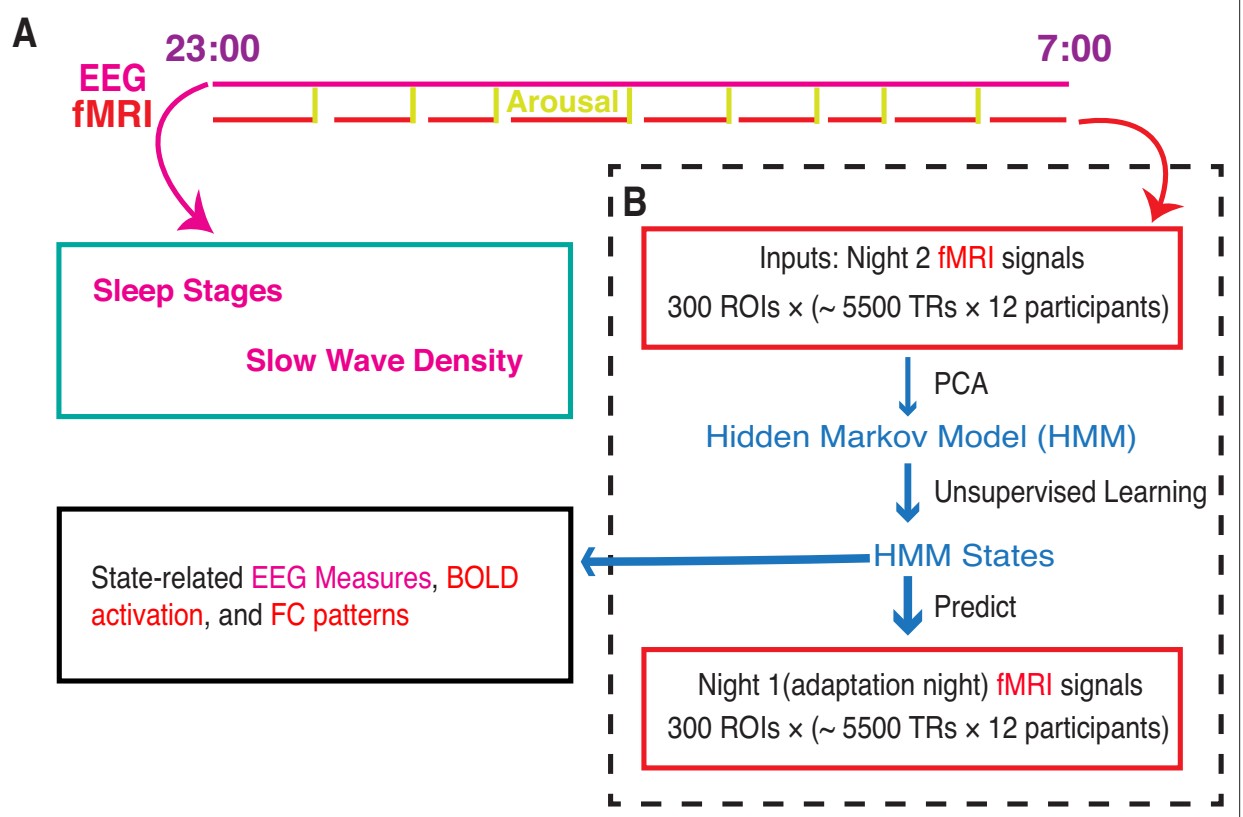

**Figure 1.** Whole-brain activity dynamic identified from functional MRI (fMRI) sleep recording using a Hidden Markov Model. (**A**) Participants slept inside a scanner from ~23:00 to ~07:00 for two consecutive nights, with concurrent EEG-fMRI recording. During each night, the fMRI experiments were intermittently disrupted by either acoustical arousals (eight random arousals) or spontaneous awakenings. Sleep stages and slow wave density were derived from EEG signals alone. (**B**) Hidden Markov model (HMM) was trained on the principal components of fMRI signals of night 2. Then the identified HMM states were generalized to night 1 fMRI signals. Finally, we studied the state-related variations in fMRI activation, FC patterns, and EEG measures. Notes: EEG, electroencephalographic; TR: repetition time; FC, functional connectivity; ROI, region of interest; PCA, principal component analysis.

The online version of this article includes the following figure supplement(s) for figure 1:

**Figure supplement 1.** Model evaluation parameters.

Furthermore, the HMM featured a transition probability matrix that detailed the likelihood of transitioning between every pair of states. Each state was accompanied by a state time course, delineating the specific time points (defined by the fMRI temporal resolution of 3 s) when the state was active. Notably, the HMM was constructed with 21 distinct states and was devoid of any prior knowledge regarding PSG staging during its estimation. For a comprehensive visual representation of the analytical process, please refer to *Figure 1* (see the Methods section for a detailed explanation). Also, there is no HMM state that was participant-specific. That is, all 21 HMM states can be found in each participant's fMRI time course.

## HMM states show PSG stage specificity

The 21 brain states (see *Figure 2B*), identified solely from fMRI, exhibited a mixture of six PSG-based sleep stages: N1, N2, N3, REM, Wake, and an 'Undefined' stage for epochs that could not be confidently assigned to one of the four following sleep stages: N1-3 and REM.

To investigate the relationship between HMM states and PSG-based sleep stages, we adopted a 'winner-takes-all' approach that assigned HMM states to the sleep stage where they most frequently occurred. Thirteen of the 21 brain states were most frequently associated with N2 sleep stages. HMM states 8 and 10 predominantly occurred during N3 sleep, while HMM states 6 and 19 were prevalent during REM sleep. HMM state 4 corresponded to the undefined sleep stage, and HMM states 13, 16, and 20 were primarily observed during Wake. Intriguingly, none of the HMM states were

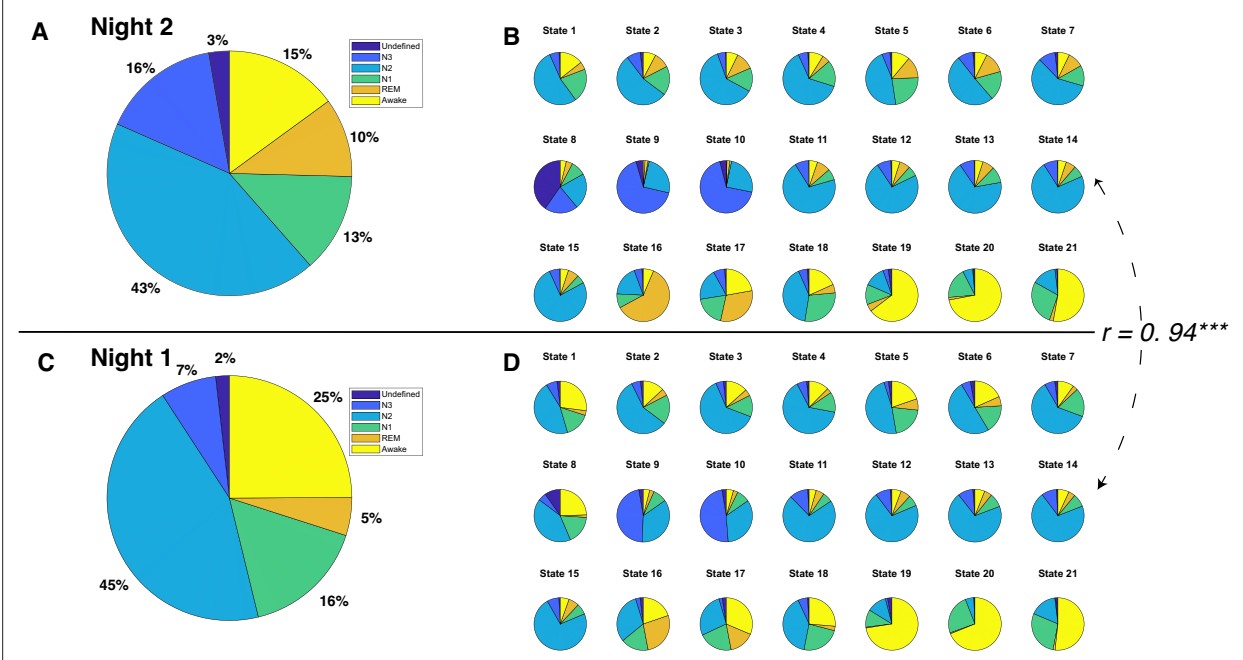

**Figure 2.** Polysomnography (PSG)-based sleep stages and Hidden Markov model (HMM) states for each night. (**A**) Distribution of sleep stages for all 12 participants during night 2. (**B**) Distribution of sleep stages for 21 HMM states during night 2. (**C**) Distribution of sleep stages for all 12 participants during night 1. (**D**) Distribution of sleep stages for 21 HMM states during night 1. The Pearson correlation coefficient between sleep stage distributions of HMM states during night 2 and those during night 1 is 0.94, p<0.0001.

The online version of this article includes the following figure supplement(s) for figure 2:

**Figure supplement 1.** Physiological variables associated with each Hidden Markov model (HMM) state during night 2.

**Figure supplement 2.** Physiological variables associated with each Hidden Markov model (HMM) state during night 1.

**Figure supplement 3.** Electroencephalography (EEG) power spectrum associated with each Hidden Markov model (HMM) state during night 2.

**Figure supplement 4.** Electroencephalography (EEG) power spectrum associated with each Hidden Markov model (HMM) state during night 1.

predominantly linked to N1 sleep. However, HMM state 11 was active for a comparable duration during both the N1 and N2 sleep stages. See *Figure 2B*.

In *Figure 3—figure supplements 1 and 2*, we plotted the time courses of two fMRI runs. A high similarity was observed between the HMM state time courses and sleep stage time courses, with the HMM time courses providing more detailed information.

The temporal characterization of these brain states enabled us to investigate the subtle details of brain dynamics within the traditional PSG-based sleep stages. The average duration, referred to as 'Lifetime,' of the HMM states varied from 8.7 to 36 s. Specifically, the mean Lifetime in states associated with N2 stages tended to be shorter compared to those linked to N3, REM, and Wake (with exceptions of state 13), as illustrated in *Figure 3—figure supplement 3*.

## Sleep states as modules of HMM state transitions

The use of a data-driven approach empowered us to explore the temporal dynamics of HMM states, and enabled us to investigate whether the fMRI-driven HMM states reveal novel dimensions of the Wake-NREM-REM-Wake sleep cycle that are hidden from traditional PSG analyses. We examined the transition probabilities among HMM states, identifying modules of HMM states that exhibited more frequent transitions between each other than to other states (*Stevner et al., 2019*; *Vidaurre et al., 2017*).

The transition probabilities of HMM brain states were organized into a 21×21 transition matrix. To explore the potential clustering of states with prevalent mutual transitions, a modularity analysis was performed on this matrix based solely on transition probabilities (see Methods section for details). As illustrated in *Figure 3*, this analysis identified five distinct transition modules, encompassing N3-, REM-, Wake-, and two different N2- modules. Importantly, this modularity analysis was conducted

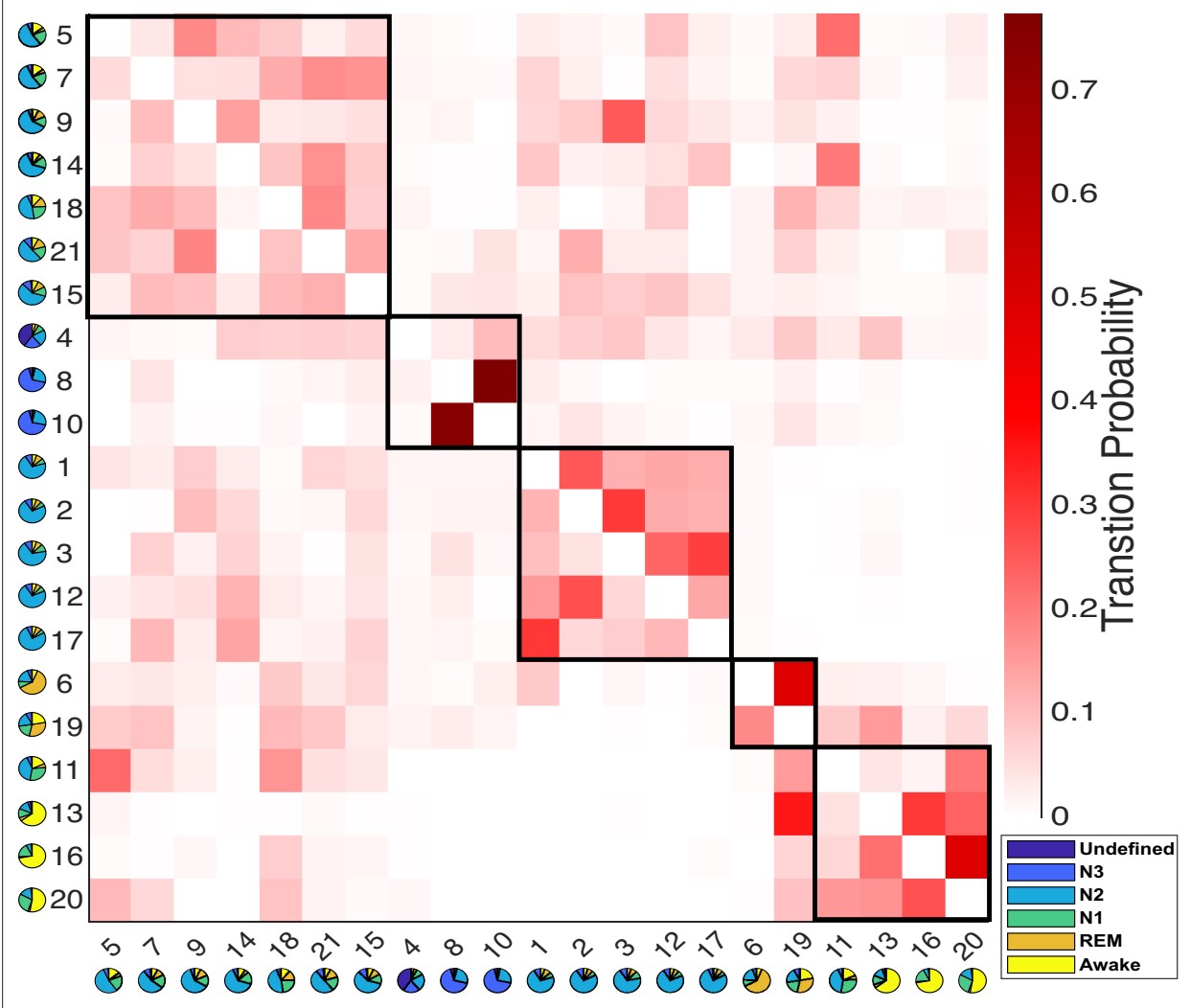

**Figure 3.** Results of the modular analysis are based solely on transition probability between Hidden Markov model (HMM) states. Each row represents the transition probability of the current HMM state (y-axis) to other states (x-axis). Twenty-one HMM states were categorized into five modules (black boxes): from left to right, light-N2 module (states 5, 7, 9, 14, 15, 18, 21), N3 module (states 4, 8, 10), deep-N2 module (states 1, 2, 3, 12, 17), rapid eye movement (REM) module (states 6 and 19), and Wake module (states 11, 13, 16, 20). The pie chart under each state represents the sleep stage distribution for the state.

The online version of this article includes the following figure supplement(s) for figure 3:

**Figure supplement 1.** State timecourse of Hidden Markov model (HMM) states and its associations with polysomnography (PSG) stages, variation in photoplethysmography (PPG) amplitude, and variations in RespRVT signals of an example run.

**Figure supplement 2.** State timecourse of Hidden Markov model (HMM) states and its associations with polysomnography (PSG) stages, variation in photoplethysmography (PPG) amplitude, and variations in RespRVT signals of a second example run.

**Figure supplement 3.** The mean Lifetime of 21 Hidden Markov model (HMM) states.

independently of PSG-based sleep stages. Interestingly, it revealed a natural clustering of states associated with the same sleep stages. For instance, two HMM states, 6 and 19, both linked to REM sleep, were grouped within the same module.

Twelve N2-related HMM states were divided into two separate modules. The first module is characterized as the light-N2 module, with higher transition probabilities to REM and Wake modules compared to the other module. The second module exhibited low transition probabilities to both the REM and Wake modules and is referred to as the deep-N2 module.

A similar duality was evident within the REM module. HMM state 19 displayed a notably higher transition probability to states in the Wake module compared to HMM state 6.

Within the Wake module, four HMM states were observed. State 11 was found to be linked to both N1 and N2 sleep stages, while the other three states (13, 16, and 20) were associated with the Wake stages. Further investigation revealed that state 13 typically occurred later in the night and later within an MRI run (see *Figure 2—figure supplement 1E, F*), suggesting it represents post-sleep wakefulness, whereas states 16 and 20 were pre-sleep wakefulness. State 13 also showed higher PPG variation, respiratory variation, and heart rates than states 16 and 20 (see *Figure 2—figure supplement 1B–D*). This observation was confirmed by the transition probability matrix, that only HMM state 13 has a lower chance of transition into N2- or N3 -related states, especially for the states within the light-N2 module, compared to HMM states 16 and 20.

## HMM states generalize to night 1 fMRI data

Next, to test the robustness of our HMM approach, we employed a semi-supervised learning approach to predict night 1 data based on the model trained on night 2 data. Specifically, we maintained state assignments from night 2 and applied the model to night 1. The resulting model indicated that despite having fewer REM and N3 stages during night 1 (See *Figure 2A, C*), there was a significant correlation between the sleep stage proportions of the HMM states for night 1 and those for night 2 ($r=0.94$, $p<0.0001$, see *Figure 2B, D*). Moreover, the physiological variables displayed similar patterns between night 1 and night 2 (see *Figure 2—figure supplements 1 and 2*).

## fMRI activation and FC patterns of HMM states

To investigate brain activity patterns specific to individual HMM states, we calculated the spatial fMRI activation map and FC pattern of each HMM state relative to the averages over all HMM states.

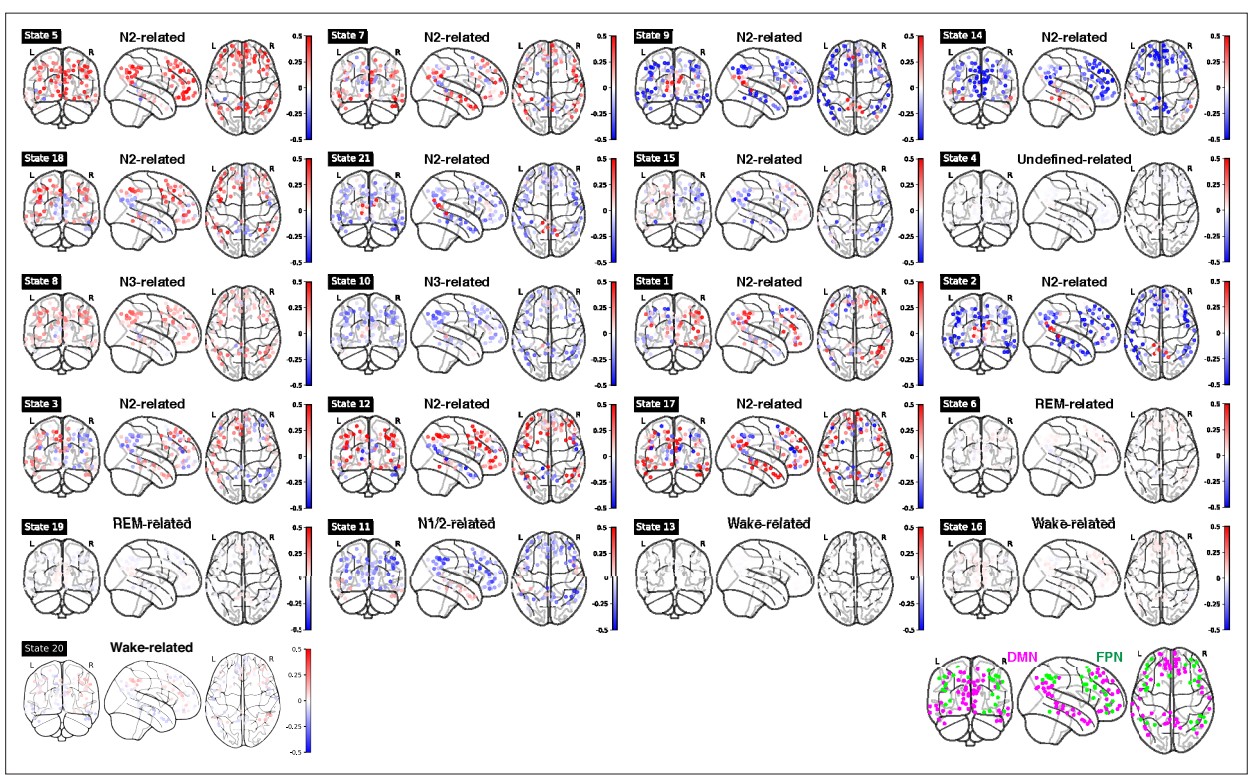

**Figure 4.** Mean functional MRI (fMRI) activation in ROIs within DMN and FPN for each Hidden Markov model (HMM) state. Bottom right panel: illustration of DMN (purple) and FPN (green) nodes. Note: DMN: Default Mode Network; FPN: Frontoparietal Network.

The online version of this article includes the following figure supplement(s) for figure 4:

**Figure supplement 1.** Mean functional MRI (fMRI) activation (percent signal change) for each state relative to baseline averaged over all Hidden Markov model (HMM) states.

**Figure supplement 2.** Functional connectivity (FC) patterns for each state relative to baseline averaged over all Hidden Markov model (HMM) states.

**Figure supplement 3.** Correlation matrix between functional connectivity (FC) patterns of each pair of Hidden Markov model (HMM) states.

*Figure 4—figure supplement 1* showcases the mean fMRI activation for each state, while the associated FC patterns are depicted in *Figure 4—figure supplement 2*.

For mean fMRI activation, Wake-related HMM state 20 demonstrated the classic opposite activation pattern between the default-mode network (DMN) and its anti-correlated networks (ACNs), see *Figure 4* and *Figure 4—figure supplement 1*. In contrast, during sleep-related HMM states, e.g., states 8 and 10, DMN and FPN showed the same activation direction.

For FC patterns, similar anti-correlated patterns were found (see *Figure 4—figure supplement 2*). In wake-related HMM states 16 and 20, the FCs between DMN and Salience Network (SAL)/ Control Network (CON) were negative, while during N3-related HMM states 8 and 10, these FCs were positive.

As expected, the FC patterns between the Visual Network (VIS) and other sensory networks (Auditory Network, AUD, and lateral/dorsal Somatomotor Network, lSMN/dSMN) were positive during wake-related HMM states but were negative during sleep-related HMM states. One notable exception was HMM state 6 (REM-related), in which VIS had a positive correlation with lSMN and AUD, mirroring those in wake-related states. During REM-related HMM states 6 and 19, the Basal Ganglia (BG) and Thalamus (THAL) had a strong positive correlation with lSMN and AUD.

When we correlated the FC patterns of each state to those of another state, the FC patterns of states that belong to the same module or are related to the same PSG-based sleep stages were highly correlated (similar to the modular results in *Figure 3*), see *Figure 4—figure supplement 3*.

## Motion parameters with sleep stages

Averaged motion across six motion parameters decreased from wake to light sleep to deep sleep at night 2. For example, the mean (standard deviation) motion for each sleep stage is as follows, N1: 0.043 (0.37); N2: 0.039 (0.033); N3: 0.035 (0.031); REM: 0.035 (0.032); Wake: 0.057 (0.052).

Similarly, the percentage of time points retained after censoring decreased from wake to light sleep to deep sleep at night 2. N1: 98.2%; N2: 99.2%; N3: 99.1%; REM: 98.7%; Wake 92.7%.

## EEG spectral features across HMM states

We conducted spectral analysis for each TR and calculated the average power spectrum of Cz for each common EEG brainwave—Delta (0.5–4 Hz), Theta (4–8 Hz), Alpha (8–13 Hz), Beta (13–30 Hz), and Gamma (30–100 Hz)—across the 21 HMM states. See *Figure 2—figure supplements 3 and 4* for night 2 and night 1 data, respectively. As expected, we found that N3-related states 8 and 10 had the highest Delta power on both nights. In addition, the Deep-N2 module had higher power in the Theta and Alpha bands compared to the Light-N2 module.

## Discussion

By applying an unsupervised learning method to night 2 of two-night fMRI sleep recordings, we deduced 21 HMM states and their transition probabilities, independently of PSG-defined sleep stages. The identified HMM states showed excellent reproducibility to night 1 data in a semi-supervised manner, a feat not previously demonstrated. Moreover, through modular analysis focused solely on transition probabilities, a duality within REM-related and N2-related HMM states was found. These findings offer unique new information about brain sleep states and their transitions that extend beyond previous PSG-based research, as well as fMRI research without whole-night recordings.

Our work addressed well-known shortcomings of PSG-based sleep staging (*Abeysuriya and Robinson, 2016*; *Decat et al., 2022*) by integrating insights from whole-brain fMRI recordings. First and foremost, while traditional PSG-based sleep staging is based on 30 s epochs, HMM analysis allows for a state-specific duration as short as the fMRI temporal resolution (here 3 s). On average, the duration of HMM states is 12 s (*Figure 3—figure supplement 3*), suggesting a more detailed characterization of brain states compared to PSG-based sleep stage analysis. Second, in terms of spatial resolution, the functional atlas used in our study encompassed 300 ROIs, offering a more detailed view of activation patterns across the entire brain, including subcortical and cerebellar regions that are ignored in PSG-based sleep staging. Third, our approach is mostly automated and objective, eliminating concerns related to inter-rater reliability issues and human error (*Lambert and Peter-Derex, 2023*; *Lee et al., 2022*; *Rosenberg and Van Hout, 2013*). Lastly, identifying transitions between

sleep stages can pose challenges when relying solely on PSG data. In contrast, the HMM is explicitly designed to model these transitions between states, providing a better understanding of the dynamic shifts that occur throughout the sleep cycle, especially when the sleep stages transition is not linear from Wake to NREM to REM in the second half of night.

Previous research suggests that the analysis of sleep data at a finer temporal resolution than PSG-based sleep staging may be valuable. For example, distinct and recurring states of waking brain activity may be as brief as 100ms during wake (*Baker et al., 2014*; *Koenig et al., 2005*). In mice, rapid (seconds-scale) fluctuations in brain-wide neuronal spiking activity have been reported during states of low alertness, attributed to fluctuation in adrenergic and cholinergic neuromodulation from basal forebrain and locus coeruleus (*Aston-Jones and Bloom, 1981*; *Collins et al., 2023*; *Kjaerby et al., 2022*; *Osorio-Forero et al., 2023*). Capturing second-scale changes of brain states with the analysis approach employed in the current study may, therefore, allow a more comprehensive investigation of the functional roles of sleep and shed light on the mechanisms by which these roles are accomplished.

The modular analysis, which solely relied on transition probabilities between states, uncovered a significant discovery. This analysis clustered HMM states into modules closely associated with PSG-defined sleep stages. This suggests that transition probabilities contain essential information about sleep states and PSG-based sleep stages. For instance, a module predominantly linked to the N3 stage consisted of two N3-related states and one undefined state. Importantly, all three states also exhibited the highest slow-wave density among all states (see *Figure 2—figure supplement 1A*).

Within the Wake module, there were four HMM states, each representing pre-sleep wake (states 16 and 20), post-sleep wake (state 13), and N1-2 (state 11). The absence of a dedicated module/state representing the N1 stage is unsurprising, considering that N1 does not distinctly manifest as a well-defined sleep stage (*Carskadon and Dement, 2011*) and it has the lowest inter-rater reliability (0.24 vs 0.76 overall) among all the PSG-defined sleep stages (*Lee et al., 2022*).

Two modules were associated with the N2 stage. One of these termed the 'Deep-N2 module,' exhibited a low likelihood of transitioning to REM and Wake while showing a slightly higher probability of transitioning to N3-related states when compared to the other module, referred to as the 'Light-N2 module.' This finding aligns with previous studies (*Brandenberger et al., 2005*; *Decat et al., 2022*), which separated the N2 stage into a quiet type (before the transition into the N3 stages, which resembles the Deep-N2 module in the current study) and an active type (preceding the transition to REM, related to the Light-N2 module).

The two REM-related states (6 and 19) within the REM module were notably different in several aspects. First, state 19 displayed a higher propensity for transitioning to the Wake module in contrast to state 6. Second, state 6 tended to occur towards the end of sleep and also late within the fMRI run (see *Figure 2—figure supplement 1E, F*). Third, in general, state 6 has a higher/stronger connections compared to state 19 (see *Figure 4—figure supplement 2*). These differences suggest an alignment of the HMM REM states along the previously defined microstates of REM, i.e., 'phasic' and 'tonic' episodes (*Simor et al., 2020*). Tonic REM is thought to be an intermediate state between wakefulness and phasic REM and is associated with a higher environmental awareness. Phasic REM occurs more often at the end of the night and is associated with a higher level of brain activity (*Simor et al., 2020*). Taken together, HMM state 19 might represent tonic REM given the high transition probability to Wake-related HMM states, while HMM state 6 might be related to phasic REM with higher FC and occurring later in the night.

In terms of both BOLD activation and FC patterns, a notable divergence between N3-related states and Wake-related states is observed in the interaction between DMN and its ACNs (SAL/CON/FPN, etc.). It is plausible that the degree of correlation or anticorrelation between DMN and its ACNs is a pivotal factor influencing the transitions from wakefulness to light sleep and, subsequently, to deep sleep. The SAL is considered crucial for cognitive control, as it handles the perception and response to homeostatic demands (*Menon, 2011*; *Peters et al., 2016*; *Seeley, 2019*). It further acts as a mediator for dynamic interactions among other prominent large-scale brain networks engaged in externally focused attention (FPN) and internally directed self-referential cognitive processes (DMN). It is plausible that during sleep, the mediating function of the SAL is temporarily suspended to allow for its restoration. Recent findings have indicated that disruptions in SAL connections were observed following one night of sleep deprivation (*Fang et al., 2015*) or in individuals with insomnia disorder (*Cheng et al., 2022*; *Li et al., 2022*; *Wei et al., 2020*).

There are a few limitations worth mentioning. First, we made an arbitrary selection of 13 principal components for PCA, accounting for 40.7% of the total variance. While this percentage of explained variance may seem low, it was a necessary step to stabilize the fitting of the HMM in the current study. Notably, the trained HMM demonstrated generalization to night 1 data, validating the chosen principal components as they encompass sufficient information about the fMRI signals. Second, while our study involved a relatively small number of participants (12), it included a large amount of fMRI data (~16 hr scan) per participant. Although the HMM trained on data from 12 participants was robust, the generalizability of the current results to different populations—such as healthy aging individuals and clinical populations—needs to be demonstrated in future studies, particularly with larger sample sizes and more diverse populations. Third, we chose to not include EEG features in our data-driven model. However, the current method is not limited to fMRI data and can be applied to EEG data. Given that previous data-driven studies based on EEG data have suggested that there might be more than five traditional sleep stages (*Christensen et al., 2019*; *Decat et al., 2022*; *Koch et al., 2014*), as well as subdivisions within these traditional sleep stages (*Brandenberger et al., 2005*; *Decat et al., 2022*; *Simor et al., 2020*), future studies may apply data-driven models on both fMRI and EEG data. Fourth, while we selected 21 HMM brain sleep states based on model evaluation parameters in the current study, the exact number of sleep states is not fixed and likely depends on various sample- and methods-related factors, such as sample size and model setups.

There are some key differences in data acquisition and analysis that make it challenging to directly compare HMM states between the current study and *Stevner et al., 2019*. First, *Stevner et al., 2019* collected only 1-hr-long sleep data from 18 participants, whereas our current study includes 8-hr-long sleep data from 12 participants for two consecutive nights. As discussed in the introduction, full sleep cycling cannot be obtained from 1 hr long sleep due to the lack of REM stage and incomplete sleep cycles. Second, in *Stevner et al., 2019*; *Figure 4e*, the four wake-NREM stages had roughly the same duration. In contrast, in our current study (night 2, *Figure 2A*), the N2 stage comprises 43% of total sleep, which aligns with the natural N2 composition of nocturnal sleep stages. This discrepancy might explain the different number of N2-related states found in the two studies, with 3 out of 19 in *Stevner et al., 2019* versus 13 out of 21 in our current study.

To summarize, we demonstrated how a data-driven analysis of an extensive sleep fMRI dataset can reproducibly characterize the full pattern of arousal state changes that recur during a whole night's sleep. The findings underscore the advantages of the whole-night fMRI data, over the traditional PSG sleep staging and previous fMRI sleep studies, in achieving a fine-grained characterization of brain sleep states and their transitions. The successful generalization of our approach trained on night 2 to night 1 data shows its robustness, reliability, and objectivity across multiple nights. Our exploration of transitions between HMM states unveiled modules closely linked to distinct sleep stages, revealing a duality within N2-related modules that further dissects N2 stages into 'light' and 'deep' N2 modules. We identified a duality with REM-related HMM states, which resembles the 'phasic' versus 'tonic' REM. Additionally, we separated pre-sleep from post-sleep Wake states. Analysis of brain activation and FC patterns of HMM states indicated that the connections between DMN and ACNs, especially SAL, may play a critical role in the transition from wake to light sleep and subsequently to deep sleep. Collectively, this enriched comprehension of brain dynamics during nocturnal sleep holds the potential for identifying promising biomarkers associated with sleep disorders that significantly impact sleep-stage transitions.

## Methods
### Data acquisition and processing

All the data used in this study followed approved human subjects research protocols approved by the National Institutes of Health Combined Neuroscience Institutional Review Board (USA, Protocol Number 16 N-0031), and informed consent was obtained from the participants. Data acquisition was conducted as part of a previously described sleep experiment (*Moehlman et al., 2019*), encompassing two consecutive nights of concurrent fMRI-EEG data collection while participants slept inside a 3T Siemens Prisma MRI scanner. To ensure a consistent sleep schedule, participants were instructed to adhere to regular sleep patterns for two weeks before the experiments, and compliance was

verified with wearable devices. No sleep deprivation protocols were implemented during the course of the study.

The fMRI data encompassed whole-brain scans consisting of 50 axial slices, captured at a spatial resolution of 2.5 mm (2.5 × 2.5 mm² in-plane), with a 2.0 mm slice thickness and a 0.5 mm slice gap. The data was acquired at a temporal resolution of 3 s, employing a 90° flip angle and an echo time of 36 ms. Data acquisition utilized a multi-slice echoplanar imaging approach in an interleaved manner. Simultaneously, EEG data was recorded at a digitization rate of 5 kHz, employing 64 channels to comprehensively cover the scalp. The MR-compatible EEG system used was from Brain Products (Gilching, Germany).

Additionally, concurrent peripheral physiological measures were acquired, including a chest belt to monitor respiratory chest excursion and finger skin photoplethysmography (PPG) to monitor cardiac rate and peripheral vascular volume. These physiological parameters were collected using a Biopac acquisition system with TSD200-MRI and TSD221-MRI transducers, combined with an MP 150 digitizer sampling at 1 kHz, sourced from Biopac in Goleta, CA, USA. To ensure accurate synchronization, data collection for EEG was timed using the 10 MHz clock from the MR instrument. The Biopac device also recorded volume triggers from the MRI scanner to facilitate synchronization of peripheral physiology recordings.

A total of 12 subjects (aged 18–35 years, including 8 females), out of 16 attempts, completed both nights of scanning (from 23:00 to 07:00 roughly). Throughout each night, the fMRI experiments were intermittently disrupted by either acoustically stimulated or spontaneous awakenings. As a consequence, a series of experimental runs was generated, with durations ranging from 5 min to 3 hr. Detailed fMRI, EEG, and peripheral physiological measures preprocessing steps can be found elsewhere (*Moehlman et al., 2019*; *Picchioni et al., 2022*). Briefly, a tailored version of the 'afni_proc' script in AFNI software was used (*Cox, 1996*), including outlier removal, detrend, RETRIOCOR (*Glover et al., 2000*), slice timing correction, motion correction, normalization, registration, global signal removal, and censoring (Euclidean Norm of the first difference of six motion parameters exceeded 0.3). Previous analysis of the same data indicated that motion during extended sleep scans is comparable to the motion observed in shorter resting-state scans (*Moehlman et al., 2019*). We also found that motion is lower during deep sleep compared to wake, see **Results**. The EEG signal underwent correction for MRI gradient and cardio-ballistic artifacts and was subsequently down-sampled to a rate of 250 Hz using the Analyzer software (Brain Vision, Morrisville, USA). The process of sleep scoring was carried out using a central electrode in 30 s epochs, in accordance with established criteria with standard filters, and channel references (*Berry et al., 2020*). ICA cleaning and slow wave auto-detection script were applied to EEG signals (*Betta et al., 2021*; *Mensen et al., 2016*; *Riedner et al., 2007*). Sleep score, slow wave density, and peripheral physiological measures were resampled into a 3 s resolution aligned with the BOLD signal.

## HMM overview

In pursuit of a data-driven approach to understanding the brain dynamics in the fMRI signals, we employed an HMM (*Vidaurre et al., 2018*; *Vidaurre et al., 2017*) to analyze timecourses extracted from 300 ROIs based on the Seitzman 300-ROI atlas (*Seitzman et al., 2020*). To prepare the data for analysis, we first standardized the participant-specific sets of 300 ROI timecourses (scaled to a mean of 0, and a standard deviation of 1), which were then concatenated across all participants. This standardization was performed separately for each night. This resulted in a data matrix with dimensions of 300 × (12 ×~5500) for each night, with approximately 5500 repetition time (TR), excluding breaks between runs and censored TR, accounting for 8 hr of scan time based on a 3 s TR.

The HMM inference process sought to find a sequence of recurring discrete states, each characterized by a distinct statistical arrangement of data. We employed a Gaussian HMM using the Matlab toolbox HMM-MAR v1.0 (https://github.com/OHBA-analysis/HMM-MAR, copy archived at *OHBA-analysis, 2024*), where each state was modeled as a multivariate normal distribution encompassing both first-order statistics (mean activity) and second-order statistics (covariance matrix). These state parameters were determined collectively at the group level, while the state timecourses were individually defined for each subject. As a result, the HMM identified periods of quasi-stationary activity, during which the 300 ROI timecourses displayed specific configurations of mean activity and FC.

Given the high spatial dimensionality of fMRI data, we employed principal component analysis (PCA) to reduce the number of parameters in the decomposition process as a common practice. This not only improves the signal-to-noise ratio but also enhances the overall robustness of HMM results (*Stevner et al., 2019*; *Vidaurre et al., 2018*; *Vidaurre et al., 2017*). By selecting the top 13 principal components, we retained 40.7% of the signal variance, resulting in a data matrix with dimensions of $13 \times (12 \times \sim 5500)$. This matrix was then input into the HMM. For a more detailed overview of the analytical workflow, please refer to *Figure 1*.

## Choice of the number of HMM states

Our analysis involved running the HMM across a range of model orders, specifically spanning from 4 to 25. The assessment of each solution encompassed various summary statistics, with the most pertinent findings illustrated in *Figure 1—figure supplement 1*.

*Figure 1—figure supplement 1A* displays the minimum free energy plotted against the HMM model order. This free energy, functioning as a statistical metric, undergoes minimization in the Bayesian optimization process, approximating the model evidence. It encapsulates two crucial factors: the model's alignment with the data and its complexity, assessed by its deviation from the prior distribution. A lower value of free energy indicates a better model. The first negative peak is observed at K=21.

To provide insights into the temporal aspects, we defined fractional occupancy as the proportion of time in which an HMM state was active. In *Figure 1—figure supplement 1B, C*, we present the evolution of maximum (median) fractional occupancy across HMM states as a function of the model order. We observe a rapid decline in this curve for low values of K, suggesting that, as anticipated, the contribution of each HMM state to the total recording time decreased with an increasing number of states. However, this trend stabilizes at approximately K=21. This phenomenon is also mirrored in the development of the mean Lifetime of the HMM state, which exhibits a similar stabilization pattern at around K=21, as indicated in *Figure 1—figure supplement 1E*.

To assess the relationship between the fMRI-based HMM states and PSG-based sleep scoring, we conducted a multivariate analysis of variance (MANOVA). The MATLAB function manova1 was employed to compute Wilk's $\Lambda$, which provides insights into how effectively the K HMM state timecourses can be categorized according to sleep scoring (the lower the better), as depicted in *Figure 1—figure supplement 1D*. There is a local minimum at K=21.

Taken together, we chose the model order K=21 as the number of HMM states. It should be noted that free energy is weighted most among those five model evaluation statistics.

## Analysis and visualization of HMM transitions

The transition probability matrix, a fundamental element explicitly modeled by HMM, exhibited a discernible structure characterized by subnetworks of HMM states that displayed more frequent transitions among themselves than to states external to their respective subnetworks. Essentially, this transition matrix could be viewed as a directed graph marked by a modular organization. This characteristic was effectively demonstrated by applying the transition matrix (depicted in *Figure 3*) to a modularity analysis. This modular analysis was performed using MATLAB functions sourced from the Brain Connectivity Toolbox (https://sites.google.com/site/bctnet/Home; *Rubinov and Sporns, 2010*), which relies on Newman's spectral community detection method (*Leicht and Newman, 2008*).

## Visualizing mean fMRI activation maps and FC patterns of HMM states

The mean distributions and covariance matrices specific to each state were subsequently projected back onto the MNI space utilizing the mixing matrix derived from the PCA. We generated mean fMRI activation maps and FC patterns for every HMM state relative to the baseline averaged over all HMM states. For FC patterns, within- or between-network connectivities were calculated as the average Fisher-transformed functional connectivity between each pair of ROIs within or between networks. For visualization purposes, we grouped 300 ROIs into 14 networks based on the Seitzman Atlas (*Seitzman et al., 2020*). In addition, we assigned subcortical and cerebellar regions to the additional four Networks: Posterior hippocampus (pHIP, anterior hippocampus is included in MTL network), basal ganglia (BG), Thalamus (THAL), and Cerebellum (CB). Hence, a total of 18 networks were used.

## Visualizing state timecourse of HMM states and its associations with PSG stages, PPG amplitude, and respiratory signals

Two example runs have been shown in *Figure 3—figure supplements 1 and 2*. These two examples showed how the HMM state timecourse (top panel) contained fine-grained information compared to the traditional PSG-based sleep stages (second panel) and also associated with PPG (third panel) and respiratory signals (last panel).

## Acknowledgements

This research was supported by the Intramural Research Program of the NIH, NINDS. Sussan Guttman, Steve Newman, and Tina T Liu are acknowledged for advice and assistance.

## Additional information

### Funding

| Funder | Grant reference number | Author |
| --- | --- | --- |
| National Institutes of Health | Intramural Research Program | Fan Nils Yang<br>Dante Picchioni<br>Jacco A de Zwart<br>Yicun Wang<br>Peter van Gelderen<br>Jeff H Duyn |

The funders had no role in study design, data collection and interpretation, or the decision to submit the work for publication.

### Author contributions

Fan Nils Yang, Conceptualization, Formal analysis, Investigation, Visualization, Methodology, Writing – original draft; Dante Picchioni, Resources, Data curation, Writing – review and editing; Jacco A de Zwart, Yicun Wang, Peter van Gelderen, Data curation, Writing – review and editing; Jeff H Duyn, Resources, Supervision, Funding acquisition, Project administration, Writing – review and editing

### Author ORCIDs

Fan Nils Yang ⓘ https://orcid.org/0000-0003-2565-6594
Jacco A de Zwart ⓘ https://orcid.org/0000-0001-8155-8185

### Ethics

All the data used in this study followed approved human subjects research protocols approved by the National Institutes of Health Combined Neuroscience Institutional Review Board (USA, Protocol Number 16-N-0031), and informed consent was obtained from the participants.

Reviewer #1 (Public review): https://doi.org/10.7554/eLife.98739.3.sa1
Reviewer #2 (Public review): https://doi.org/10.7554/eLife.98739.3.sa2
Author response https://doi.org/10.7554/eLife.98739.3.sa3

## Additional files

### Supplementary files
• MDAR checklist

### Data availability

The datasets are available at https://openneuro.org/datasets/ds005127/versions/1.0.2. The codes are available at https://github.com/nilsyang/Codes, copy archived at *Yang, 2024*.

The following dataset was generated:

| Author(s) | Year | Dataset title | Dataset URL | Database and Identifier |
|---|---|---|---|---|
| Picchioni D, Duyn JH, de Zwart JA | 2024 | AMRI 16-N-0031 sleep1 | https://openneuro.org/datasets/ds005127/versions/1.0.2 | OpenNeuro, ds005127 |

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
