## [Editor Report · eLife assessment]

This **important** work, leveraging state-of-the-art whole-night sleep EEG-fMRI methods, advances our understanding of the brain states underlying sleep and wakefulness. Despite a small sample size, the authors present **convincing** evidence for substates within N2 and REM sleep stages, with reliable transition structure, supporting the perspective that there are more than the five canonical sleep/wake states.

---

## [Referee Report · Reviewer #1 (Public review)]

Summary:

The study made fundamental findings in investigations of the dynamic functional states during sleep. Twenty-one HMM states were revealed from the fMRI data, surpassing the number of EEG-defined sleep stages, which can define sub-states of N2 and REM. Importantly, these findings were reproducible over two nights, shedding new light on the dynamics of brain function during sleep.

Strengths:

The study provides the most compelling evidence on the sub-states of both REM and N2 sleep. Moreover, they showed these findings on dynamics states and their transitions were reproducible over two nights of sleep. These novel findings offered unique information in the field of sleep neuroimaging.

Comments on revised version:

Nice work! All my concerns have been addressed, and I have no further suggestions.

---

## [Referee Report · Reviewer #2 (Public review)]

Summary:

Yang and colleagues used a Hidden Markov Model (HMM) on whole-night fMRI to isolate sleep and wake brain states in a data-driven fashion. They identify more brain states (21) than the five sleep/wake stages described in conventional PSG-based sleep staging, show that the identified brain states are stable across nights, and characterize the brain states in terms of which networks they primarily engage.

Strengths:

This work's primary strengths are its dataset of two nights of whole-night concurrent EEG-fMRI (including REM sleep), and its sound methodology.

Weaknesses:

Weaknesses are its small sample size, and limited attempts at relating the identified fMRI brain states back to EEG.

General appraisal:

The paper's conclusions are generally well-supported, but additional analyses could improve the work further.

The authors' main focus lies in identifying fMRI-based brain states, and they succeed at demonstrating both the presence and robustness of these states in terms of cross-night stability. Additional characterization of brain states in terms of which networks these brain states primarily engage adds additional insights.

A missed opportunity remains the absence of more analyses relating the HMM states back to EEG. While the authors show how power in different EEG bands varies with HMM state (Supplementary Figures 10 and 11) it would be much more informative to show the complete EEG spectra for each of the 21 HMM states, organized by PSG-based sleep/wake state. This would enable answering how EEG spectra of, say, different N2-related HMM states compare. Similarly, it is presently unclear whether anything noticeable happens within the EEG timecourse at the moment of an HMM class switch (particularly when the PSG stage remains stable). Such analyses might have shown that fMRI-based brain states map onto familiar EEG substates, or reveal novel EEG changes that have so far gone unnoticed. Furthermore, if band-specific analyses are to be performed, care should be taken to specify bands in accordance with the dominant sleep EEG features (e.g., slow oscillation and sigma/spindle bands are currently missing).

---

## [Author Response]

The following is the authors’ response to the original reviews.

**Reviewer #1 (Public Review):**
Summary:The study made fundamental findings in investigations of the dynamic functional states during sleep. Twenty-one HMM states were revealed from the fMRI data, surpassing the number of EEG-defined sleep stages, which can define sub-states of N2 and REM. Importantly, these findings were reproducible over two nights, shedding new light on the dynamics of brain function during sleep.Strengths:The study provides the most compelling evidence on the sub-states of both REM and N2 sleep. Moreover, they showed these findings on dynamics states and their transitions were reproducible over two nights of sleep. These novel findings offered unique information in the field of sleep neuroimaging.Weaknesses:The only weakness of this study has been acknowledged by the authors: limited sample size.

We thank the reviewer for the overall enthusiasm for this study.

**Reviewer #1 (Recommendations For The Authors):**
(1) Were there differences in the extent of head motion during sleep among sleep stages? How was the potential motion parameter differences handled during the statistical analyses?If there were large head motions that continued for a long time (e.g., longer than 1 minute), how did the authors deal with that scanning session? For an extremely long scanning session (3 hours), how was motion correction conducted? It would be great if the authors could provide more details.

We found that N3 sleep stage had lowest head motion, followed by REM, N2, N1, and lastly Wake. In other words, participants have lower head motion during sleep than during Wakefulness. We added this information to the Supplemental Results, copied below.

We performed standardized motion correction during preprocessing using AFNI regardless of the duration of the scans. We did not include motion parameters in the HMM model. Time frames with Excessive head motion (any of 6 head motion parameters exceeding 0.3 mm or degree) was censored. Previous analysis of the same data indicated that motion during extended sleep scans is comparable to the motion observed in shorter resting-state scans (Moehlman et al., 2019).

In Supplemental Results, “Motion parameters with sleep stages.

Averaged motion across six motion parameters decreased from wake to light sleep to deep sleep at night 2. For example, the mean (standard deviation) motion for each sleep stage is as follows, N1: 0.043 (0.37); N2: 0.039 (0.033); N3: 0.035 (0.031); REM: 0.035 (0.032); Wake: 0.057 (0.052).

Similarly, the percentage of time points retained after censoring decreased from wake to light sleep to deep sleep at night 2. N1: 98.2%; N2: 99.2%; N3: 99.1%; REM: 98.7%; Wake 92.7%.

In the method section, “Previous analysis of the same data indicated that motion during extended sleep scans is comparable to the motion observed in shorter resting-state scans (Moehlman et al., 2019). We also found that motion is lower during deep sleep compared to wake, see Supplemental Results.”

(2) It is possible that the data input for the HMM analyses might vary among participants and between the two nights, how did the authors deal with this issue during statistical analyses?

This is a great question. We standardized BOLD timecourses for each participant and each night to avoid differences among participants and between two nights. We revised the description in the method section to make this point clear.

In the method section, “To prepare the data for analysis, we first standardized the participant-specific sets of 300 ROI timecourses (scaled to a mean of 0, and a standard deviation of 1), which were then concatenated across all participants. This standardization was performed separately for each night. ”

(3) Figures 2 and 4, the top part seems to be missing, e.g., "Night 2" in Figure 2, and "N2-related" in Figure 4.

Thank you for pointing out these errors. We fixed them.

(4) Figure 3 seems to be more stretched vertically than horizontally.

We revised the figure to ensure it appears balanced on both sides.

**Reviewer #2 (Public Review):**
Summary:Yang and colleagues used a Hidden Markov Model (HMM) on whole-night fMRI to isolate sleep and wake brain states in a data-driven fashion. They identify more brain states (21) than the five sleep/wake stages described in conventional PSG-based sleep staging, show that the identified brain states are stable across nights, and characterize the brain states in terms of which networks they primarily engage.Strengths:This work's primary strengths are its dataset of two nights of whole-night concurrent EEG-fMRI (including REM sleep), and its sound methodology.Weaknesses:The study's weaknesses are its small sample size and the limited attempts at relating the identified fMRI brain states back to EEG.

We thank the reviewer for the positive feedback and helpful suggestions for this study.

General appraisal:The paper's conclusions are generally well-supported, but some additional analyses and discussions could improve the work.The authors' main focus lies in identifying fMRI-based brain states, and they succeed at demonstrating both the presence and robustness of these states in terms of cross-night stability. Additional characterization of brain states in terms of which networks these brain states primarily engage adds additional insights.A somewhat missed opportunity is the absence of more analyses relating the HMM states back to EEG. It would be very helpful to the sleep field to see how EEG spectra of, say, different N2-related HMM states compare. Similarly, it is presently unclear whether anything noticeable happens within the EEG time course at the moment of an HMM class switch (particularly when the PSG stage remains stable). While the authors did look at slow wave density and various physiological signals in different HMM states, a characterization of the EEG itself in terms of spectral features is missing. Such analyses might have shown that fMRI-based brain states map onto familiar EEG substates, or reveal novel EEG changes that have so far gone unnoticed.

We thank the reviewer for this great suggestion. We performed EEG spectral analysis on each HMM state. Results were added to Suppementary Results and Supplementary Figure 10 and 11 (Copied below). Specifically, we confirmed that N3-related states had highest Delta power and that the Deep-N2 module showed different spectral profiles compared to Light-N2 module. Unfortuantely, we could not perform EEG analysis at the moment of an HMM class switch, given that there are too many different type of HMM switches (21*20/2).

In Supplemental Results: “We conducted spectral analysis for each TR and calculated the average power spectrum for each common EEG brainwave—Delta (0.5-4 Hz), Theta (4-8 Hz), Alpha (8-13 Hz), Beta (13-30 Hz), and Gamma (30-100 Hz)—across the 21 HMM states. See Supplementary Figure 10 and 11 for night 2 and night 1 data, respectively. As expected, we found that N3-related states 8 and 10 had highest Delta power in both nights. In addition, the Deep-N2 module had higher power in Theta and Alpha bands compared to the Light-N2 module.”

It is unclear how the presently identified HMM brain states relate to the previously identified NREM and wake states by Stevner et al. (2019), who used a roughly similar approach. This is important, as similar brain states across studies would suggest reproducibility, whereas large discrepancies could indicate a large dependence on particular methods and/or the sample (also see later point regarding generalizability).

This is a great question. There are some similarities and differences between the current study and Stevner et al. (2019). We discussed this in the Supplementary Discussion. Copied below.

In the Supplementary Discussion: “Both studies demonstrated that HMM states can be effectively divided into meaningful modules solely based on transition probabilities. Furthermore, both studies indicated that pre-sleep wakefulness differs from post-sleep wakefulness.

However, despite the similar approaches used, key differences in data acquisition and analysis make it challenging to directly compare HMM states between these two studies. Firstly, Stevner et al. (2019) collected only 1-hour-long sleep data from 18 participants, whereas our current study includes 8-hour-long sleep data from 12 participants for two consecutive nights. As discussed in the main text, full sleep cycling cannot be obtained from 1-hour long sleep due to the lack of REM stage and incomplete sleep cycles. Secondly, in Stevner et al. (2019) (Figure 4e), the four wake-NREM stages had roughly the same duration. In contrast, in our current study (Night 2, Figure 2A), the N2 stage comprises 43% of total sleep, which aligns with the natural N2 composition of nocturnal sleep stages. This discrepancy might explain the different number of N2-related states found in the two studies, with 3 out of 19 in Stevner et al. (2019) versus 13 out of 21 in our current study.”

More justice could be done to previous EEG-based efforts moving beyond conventional AASM-defined sleep/wake states. Various EEG studies performed data-driven clustering of brain states, typically indicating more than 5 traditional brain states (e.g., Koch et al. 2014, Christensen et al. 2019, Decat. et al 2022). Beyond that, countless subdivisions of classical sleep stages have been proposed (e.g., phasic/tonic REM, N2 with/without spindles, N3 with global/local slow waves, cyclic alternating patterns, and many more). While these aren't incorporated into standard sleep stage classification, the current manuscript could be misinterpreted to suggest that improved/data-driven classifications cannot be achieved from EEG, which is incorrect.

We agree with the reviewer that previous EEG-based efforts should be mentioned. We now added this in the manuscript. Copied below.

In the Discussion section, “Third, we chose to not include EEG features in our data-driven model. However, the current method is not limited to fMRI data and can be applied to EEG data. Given that previous data-driven studies based on EEG data have suggested that there might be more than five traditional sleep stages (Christensen et al., 2019; Decat et al., 2022; Koch et al., 2014), as well as subdivisions within these traditional sleep stages (Brandenberger et al., 2005; Decat et al., 2022; Simor et al., 2020), future studies may apply data-driven models on both fMRI and EEG data. ”

More discussion of the limitations of the current sample and generalizability would be helpful. A sample of N=12 is no doubt impressive for two nights of concurrent whole-night EEG-fMRI. Still, any data-driven approach can only capture the brain states that are present in the sample, and 12 individuals are unlikely to express all brain states present in the population of young healthy individuals. Add to that all the potentially different or altered brain states that come with healthy ageing, other demographic variables, and numerous clinical disorders. How do the authors expect their results to change with larger samples and/or varying these factors? Perhaps most importantly, I think it's important to mention that the particular number of identified brain states (here 21, and e.g. 19 in Stevner) is not set in stone and will likely vary as a function of many sample- and methods-related factors.

We thank the reviewer for the great suggestions. We now included these points when discussing limitations in the Discussion section. We think that a HMM model with larger sample size might produce more fine-grained results, but this remains to be investigated when a more extensive dataset becomes available.

In the Discussion section, “Secondly, while our study involved a relatively small number of participants (12), it included a large amount of fMRI data (~16 hours scan) per participant. Although the HMM trained on data from 12 participants was robust, the generalizability of the current results to different populations—such as healthy aging individuals and clinical populations—needs to be demonstrated in future studies, particularly with larger sample sizes and more diverse populations.”

“Fourth, while we selected 21 HMM brain sleep states based on model evaluation parameters in the current study, the exact number of sleep states is not fixed and likely depends on various sample- and methods-related factors, such as sample size and model setups.”